# Selection of Flagship Species and Their Use as Umbrellas in Bird Conservation: A Case Study in Lishui, Zhejiang Province, China

**DOI:** 10.3390/ani13111825

**Published:** 2023-05-31

**Authors:** Yifei Wang, Jie Shi, Yi Wu, Wenwen Zhang, Xiao Yang, Huanxin Lv, Shaoxia Xia, Shengjun Zhao, Jing Tian, Peng Cui, Jiliang Xu

**Affiliations:** 1School of Ecology and Nature Conservation, Beijing Forestry University, 35 East Qinghua Road Haidian District, Beijing 100083, China; w1170819038@163.com; 2State Environmental Protection Key Laboratory on Biodiversity and Biosafety, Research Center for Biodiversity Conservation and Biosafety, Nanjing Institute of Environmental Sciences, Ministry of Ecology and Environment, Nanjing 210042, China; shijie.ahu@gmail.com (J.S.); wuyi@nies.org (Y.W.); zhangww1023@126.com (W.Z.); yx2425471650@163.com (X.Y.); zhaoshengjun1216@126.com (S.Z.); tianjingjing444@163.com (J.T.); 3Key Laboratory of Ecosystem Network Observation and Modeling, Institute of Geographic Sciences and Natural Resources Research, Chinese Academy of Sciences, Beijing 100101, China; lvhuanxin77@163.com (H.L.); xiasx@igsnrr.ac.cn (S.X.)

**Keywords:** flagship species, analytic hierarchy process, entropy weight method, MaxEnt model, umbrella species

## Abstract

**Simple Summary:**

The concept of flagship species plays a vital role in biodiversity conservation. In order to establish a technical route for selecting flagship species and strengthen the application of the concept of flagship species in biodiversity conservation at the local scale, we selected birds as a target group and conducted a study on selecting flagship species in Lishui, Zhejiang Province, China. Through the analytic hierarchy process-entropy weight method (AHP-EM) and the MaxEnt model, 10 bird species were selected as the flagship species of Lishui, and a further analysis showed that these 10 species were fully representative of birds of Lishui and that their distribution covered the main protected areas in Lishui. It is hoped that the selection method of flagship species detailed in this study can provide a reference and promote a biodiversity conservation level for other regions.

**Abstract:**

The concept of flagship species is widely used in conservation biology. Flagship birds play a key role in raising conservation funds, increasing awareness of biodiversity conservation, and maintaining ecosystem services. This study selected flagship bird species in Lishui, Zhejiang Province, China, and assessed their conservation effectiveness and ability to serve as umbrella species. A regional bird survey program from 2019–2022 recorded 361 bird species in Lishui. This study constructed a framework of flagship species selection based on social, ecological, economic, and cultural criteria. The analytic hierarchy process-entropy weight method (AHP-EM) was used to rank the score of 361 bird species, and the MaxEnt model was used to analyze the suitable distribution areas of these species. Finally, 10 species, which covered the distribution sites of all 361 bird species, were selected as the flagship species of Lishui. The distribution areas covered all the nature reserves and the priority areas of biodiversity of Lishui, in which these 10 species can also serve as umbrella species to protect local biodiversity. The methodology and ideas in this study could provide insights into the application of conservation concepts at the local level, as well as suggest possible recommendations for local governments to select flagship species for conservation.

## 1. Introduction

The Earth is currently experiencing its sixth mass extinction event [1,2] due to habitat destruction, over-exploitation, environmental change, climate change, and other reasons [3]. One of the most important challenges facing humanity in the 21st century is how to prevent species extinction and natural hazards, and the conservation of biodiversity has become a topic of concern around the world [4]. Successful biodiversity conservation requires stable and reliable financial support [5]. The Resource Mobilization Expert Group at the 15th Conference of the Parties to the United Nations Convention on Biological Diversity (CBD) estimated that global funding requirements to meet the targets of the Kunming-Montreal global biodiversity framework range from USD 150 billion to USD 300 billion per year, with only around USD 78–91 billion currently spent annually on biodiversity worldwide [6]. In the case of limited capital supply and protection, it is essential to choose a cost-effective way to use all available resources [7]. To increase the amount of biodiversity that can be conserved with a limited budget, scientists have developed several concepts to try to educate the public on the problem of biodiversity loss and identify species of potential conservation value [8,9]. Among these approaches, the use of flagship species and umbrella species has been an effective and widely used concept for the restoration and conservation of biodiversity across the globe [10].

In this context, the positive impacts of flagship species have been intensively discussed and applied [11]. However, the appropriateness and validity of the flagship species concept are highly debated [12]. The selection of flagship species appears to be based purely on their marketing value, with no correlation to their endangerment status or their status and value in the ecosystem [12,13]. For flagship species to better act as ‘ambassadors’ [14], many scientists have adjusted the selection methods and criteria for flagship species: flagship species should not only have strong spiritual, popular aesthetic, or social and cultural connotations based on the economic, cultural, historical, and social values of different species in different countries or regions [15], but they should also have broader ecological and economic conservation value [16]. The use of flagship species with charisma and appeal as an important conservation tool to mobilize positive emotions and attitudes linked to interest in conservation action can help to promote and control the management of large areas of habitat and protect many other lesser-known species [17,18]. Umbrella species can protect numerous co-occurring species, share similar habitat criteria (overlapping ecological niches) or interact with each other [19,20,21], and have an ecological role that distinguishes them from flagship species. Conservationists have proposed the concept of a flagship umbrella species, where a species functions as both a flagship species and an umbrella species. The giant panda (*Ailuropoda melanoleuca*) in China and the Bale monkey (*Chlorocebus djamdjamensis*) in Ethiopia are good examples [9,10,22,23].

Birds are often used as flagship species to gain support for conservation, including in public relations, education, and fundraising. This is due to the variety of shapes, colors, and sounds that highlight the beauty of birds and the ease with which they can be found and distinguished [24]. Within the 27 Brazilian federal units, birds are most often used as regional flagship species, with all Brazilian states having at least one bird as their flagship species [15]; Hornbills (*Bucerotidae*) [25] and White-tailed Eagles (*Haliaeetus albicilla*) [26] are good examples of species that make important contributions to biodiversity conservation as flagship species. Birdwatching tourism [27,28,29], as a typical ecotourism type, is considered one of the fastest-growing nature-based tourism industries in the world [28].

As a prefecture-level city in Zhejiang Province, China, Lishui is one of seventeen key areas of global significance for biodiversity conservation in mainland China. Lishui preserves a typical and intact central subtropical forest ecosystem in eastern China, with a rare and large area of broad-leaved evergreen forest zonal vegetation. The species diversity of Lishui is the highest in Zhejiang province, which is known as the biological kingdom of east China, containing endemic species such as *Abies beshanzuensis* (Pinaceae) [30] and the Baishanzu Horned Toad (*Megophrys baishanzuensis*) [31]. A recent botanical survey revealed the presence of 42 species of rare and endangered plants among the wild higher plants in Lishui, accounting for 76.4% of the total number of rare and endangered plants in the province. Although the public plays a vital role in biodiversity conservation, flora and fauna are unfamiliar to the public and it is not easy to raise public awareness and understanding of them in a short period, making the task challenging. Lishui is located in the Wuyi Mountains Priority Area, one of China’s priority areas for biodiversity conservation. With its rich bird diversity and good ecological environment, Lishui has strong natural resources for the development of birdwatching tourism [32].

This study selected birds as the target species based on the results of a background survey on bird diversity in Lishui. In this study, we selected flagship bird species in Lishui by using the analytic hierarchy process (AHP) [33], the entropy weight method (EWM) [34], and the analytic hierarchy process-entropy weight method (AHP-EWM) [35], and we also testing the feasibility of using these species as umbrella species by using the MaxEnt model [36].

## 2. Materials and Methods

### 2.1. Study Area

Lishui is located in the southwest of Zhejiang Province, China (118°41′–120°26′ E, 27°25′–28°57′ N), with a total area of 17,298 km^2^ [37] (Figure 1). Lishui is a prefecture-level city with the largest land area in Zhejiang Province. The region is in the subtropical monsoon climate zone, with a mild climate, warm winters and early springs, long frost-free periods, and abundant rainfall, with an annual average temperature of 17.8 °C. The terrain is dominated by mid-mountain and hilly landforms, sloping from the southwest to the northeast. Lishui is rich in bird biodiversity and provides habitat for nine species of Class I state key protected wild birds, including Cabot’s Tragopan (*Tragopan caboti*) and Elliot’s Pheasant (*Syrmaticus ellioti*), and 65 species of Class II state key protected wild birds. The region has the highest species diversity in Zhejiang Province and is important for biology study [38].

### 2.2. Bird Distribution Data

The bird distribution data were derived from a background survey of bird diversity in Lishui. The survey was conducted from December 2019 to February 2022. A total of 258 line transects and 112 point counts were laid out into 183 grids, each 10 × 10 km in size. Each line transect was at least 1 km, with a total line transect length of 1238 km (Figure 1). Each transect and point was surveyed in each of the four seasons in a year. Grids with multiple line transects and point counts existed, and survey data were statistically categorized to record all bird species surveyed for that grid. The observed species were identified using A Field Guide to the Birds of China [39] and classified according to the List of Birds Classification and Distribution in China (third edition) [40].

### 2.3. Environmental Data

In this study, twenty-seven environmental variables influencing the bird species distribution were selected for the initial simulation of the model from five aspects, i.e., climate, vegetation, topography, man-made interference, and land utilization (Table 1). The 27 environmental variables were cropped according to the study area boundaries and resampled to a spatial resolution of 30 × 30 m, and the coordinate system was standardized to WGS_1984_UTM_zone_51N, thus ensuring that the boundaries and ranks of each environmental factor were consistent. Finally, the processed raster variables were converted to ASCII format.

### 2.4. Selection of Flagship Species

#### 2.4.1. Ranking of Birds

The analytic hierarchy process-entropy weight method (AHP-EWM) was used to rank the birds of Lishui. To determine the weights accurately, this study used a combination of subjective and objective methods to determine the indicator weights in the flagship species evaluation system for birds. The indicator weights were first calculated separately using the AHP method and the EWM. The weights of the two methods were then combined using the principle of minimum information entropy, and the least-squares method was used to optimize the weight model [41].

As each indicator had different units, dimensions, and orders of magnitude, to unify the standards, all evaluation indicators were first standardized and transformed into standard values with no dimensions or differences in order of magnitude. Scores were assigned in the range of 0–1 before being analyzed and evaluated. This study identified nine elements that influenced the selection of flagship species and specified the ranking and scoring criteria for each element (Table 2). These criteria and the determination of scores were based on the combined opinions of experts and authors in the relevant fields rather than social surveys. The individual and combined indicator weights with the value *y_j_* were synthesized from the dimensionless processing of the indicators for each target species to obtain the assessed values of the subjective and objective indicators and the combined assessed value *Z_j_* for each target species. The combined assessed values for the target species were then ranked.
(1)$$z_{j}=\sum_{j=1}^{m}w_{j}y_{j}$$

#### 2.4.2. Potential Distribution of Species

The MaxEnt model was used to predict the potential distribution of species. Many bioclimatic variables were spatially auto-correlated, which could lead to the over-fitting of model predictions. The selection and correlation testing of all environmental variables was required to improve the accuracy of the model predictions [42]. The jackknife procedure was used to analyze the relative degree of influence of the environmental variables on the potential distribution of the species [43]. The area under the curve (AUC) of the receiver operating characteristics (ROC) curve was used to verify the predictive accuracy of the model simulation results. The AUC was used as a test of the model’s prediction accuracy, with values closer to 1.0 indicating a higher degree of predictive accuracy. The evaluation criteria were as follows: poor at 0.6–0.7, fair at 0.7–0.8, good at 0.8–0.9, and excellent at 0.9–1.0 [44]. The maximum training sensitivity plus the specificity threshold was chosen to convert the 0–1 continuous species potential distribution predictions into a binary distribution of 0 or 1, where 1 represented the presence of a species and 0 indicated no distribution [45].

#### 2.4.3. Identifying the Flagship Species

The results of the MaxEnt distribution model developed for the flagship species were superimposed on the distribution patterns of ‘all birds’ (the 361 species modelled) to determine whether the suitable range of the flagship species included other bird sites.

### 2.5. Protection Effectiveness of Flagship Species

An overlay analysis was used to assess the protection effectiveness of the flagship species. Suitable distribution areas for the flagship species were overlaid with nature reserves and priority areas for biodiversity conservation, and the overlaid distribution was then analyzed for vacancies with priority areas and reserves. This was also used to test whether the selected flagship species could also take on the function of umbrella species protection.

## 3. Result

### 3.1. Bird Species Diversity

A total of 361 species were recorded in this study, and they belonged to 19 orders and 80 families. Of these, the largest number of species was recorded in the order Passeriformes, with 184 species in 39 families, accounting for 48.75% of the total number of families and 50.97% of the total number of species. The next largest number of families was nine families of Charadriiformes, accounting for 11.25% of the total number of families. Residents accounted for 38.50%, passage migrants accounted for 26.87%, winter birds accounted for 18.28%, and summer-breeding birds accounted for 16.34%.

The survey recorded nine species of Class I state key protected wild bird, namely Cabot’s Tragopan, Elliot’s Pheasant, Baer’s Pochard (*Aythya baeri*), the Scaly-sided Merganser (*Mergus squamatus*), the Siberian Crane (*Leucogeranus leucogeranus*), the Oriental Stork (*Ciconia boyciana*), the Black-faced Spoonbill (*Platalea minor*), the White-eared Bight Heron (*Gorsachius magnificus*), and the Yellow-breasted Bunting (*Emberiza aureola*). Sixty-five species of the wild birds (18.01% of the total) identified are protected as Class II state key protected wild birds. According to China’s Red List of Biodiversity includes: vertebrates [46], three species of which were recorded at the critically endangered level; six species at the endangered level; eleven species at the vulnerable level; and forty-three species at the near threatened level.

### 3.2. Selection of Flagship Species

The scores of the 361 bird species were calculated using the AHP, EWM, and AHP-EWM methods. The target species were ranked according to the AHP-EWM scores. The top 50 species ranked according to the combined method are shown in Table 3.

To achieve effective umbrella protection, potential species should have a sufficient habitat width to cover most of the space in the habitat of each target species within their range [21]. The potential species were predicted using the MaxEnt model to determine suitable ranges and perform an overlay analysis with the full range of birds.

In terms of prediction accuracy, the MaxEnt model performed well. The AUC values of Yellow-Breasted bunting, Scaly-sided Merganser, Cabot’s Tragopan, Mandarin Duck, White-necklaced Partridge, and Elliot’s Pheasant were all >0.9, with excellent prediction accuracy being achieved. The study showed that the prediction results of all the models were highly reliable, and there was a good correlation between the environmental variables selected for each species and the prediction results, which could be used for the habitat suitability assessment of the species (Figure 2 and Table 4).

The results showed that the suitable distribution area obtained by superimposing the potential distribution maps of the top 10 species covered the distribution sites of all birds. Therefore, this study selected the top 10 species as flagship species (Figure 3).

In terms of ecotype, among the ten bird species selected using the above method, four species of land birds were represented by rare pheasants, there were two species each of songbirds and swimming birds, and one species each of raptors and climbing birds. In terms of the IUCN Red List of Threatened Species [47], there were six threatened species at the VU level and above. In terms of conservation status, all the species selected were key protected wild animals in China. In terms of residence type, seven species were mainly resident birds (Table 5).

### 3.3. Overlap Analysis of the Protection Areas

As a prefecture-level city in Zhejiang Province, China, Lishui has the designated Baishanzu National Park, five nature reserves at or above the provincial level, and seventeen biodiversity conservation priority areas. The results showed that the potential distribution areas of the 10 flagship species covered 74.70% of the total area of Lishui, 84.62% of the biodiversity conservation priority areas, and 94.65% of the nature reserves (including the national park) (Figure 4).

## 4. Discussion

### 4.1. Improvement in Flagship Species Selection Method

In recent years, a variety of approaches have been used to select flagship species. For example, Veríssimo emphasized the marketing role of flagship species and an interdisciplinary framework to improve flagship identification [12]. Roll developed a selection method for interest in reptiles using Wikipedia page viewing rates [48]. Qian established six criteria based on conservation biology, ecosystem function, and socioeconomic and cultural importance, and a candidate flagship species was identified if it satisfied four of the six indicators [18]. The flagship species selected using these methods all seem to contribute to local conservation activities, but it is worth noting that most of the selection methods are applicable to large-scale areas, lacking methodological references at smaller scales (municipal and county), as well as the in-depth confirmation of whether flagship species have any ecological value as umbrella species.

#### 4.1.1. Combined Subjective and Objective Weighting Analysis

In the selection criteria for the weighting system, this study chose a variety of criteria for the selection of the flagship species, some of which were related to the priority given to the ecological value of the species (the conservation rank of the target species, endangerment rank, stability of population size and distribution, population size, and knowability of distribution) and the ecological uniqueness of the species, so that the flagship species selected would have a significant umbrella effect and would be effective in protecting other species in the ecosystem. Other criteria were related to the role of a species in society, the economy, and culture, which can provide a variety of services such as sources of inspiration, aesthetics, and education and play an important role in forming a sense of place, social values, and cultural identity [49], as well as representing the cultural heritage of each region or country [50].

The previous selection method based on using the AHP alone to determine indicator weights was improved [51]. As a subjective weighting method, the AHP fully considers the knowledge, experience, and intention preferences of decision makers. Although the ranking of index weights often has a high rationality, it cannot overcome the shortcomings of subjective randomness. As an objective weighting rule, the EWM fully excavates the information contained in the original data itself and determines the index weight completely according to the attributes of the index value. However, the EWM cannot reflect the opinions of decision makers, and the weights obtained may not be consistent with the actual importance or can even contradict it [52]. The use of EWM and AHP alone cannot reflect both the actual importance and significance of the indicators. In the AHP-EWM method, the AHP is combined with the EWM, and least-squares combination optimization is performed [41]. This method not only reduces the subjective influence of the AHP but also weakens the degree of deviation between some conclusions of the EWM and the actual situation, making the results more scientific [53].

In this study, although the order of the 10 flagship species in the EWM and AHP-EWM rankings differed, the same set of flagship species ranked in the top 10. However, in the AHP scoring ranking results, the common kestrel was ranked ninth, while the flagship species mandarin duck was ranked seventeenth. Comparing the scoring results of the two methods revealed a large difference between criterion 3 and criterion 9. For selection criterion 3, ‘Stability of population size and distribution’, the common kestrel population size and distribution in Lishui were relatively stable and common, scoring 1.0 point, while the mandarin duck population size and distribution were relatively stable and rare, scoring 0.6 points. For criterion 9, ‘Recognition’, the mandarin duck was well known to the public, scoring 1.0, while the common kestrel was almost exclusively known to professional bird researchers or bird enthusiasts, scoring 0.1. The weight of criterion 3 in the AHP was much greater than the weight of criterion 9; as such, the common kestrel ranked ahead of the mandarin duck. In the EWM, if the information entropy *E_j_* of an indicator is smaller, the degree of variation in the value of the indicator is greater and more information is provided; moreover, the role that *E_j_* can play in the comprehensive evaluation as well as the weight becomes greater [54]. The *E_j_* = 0.892 for criterion 3 was greater than the *E_j_* = 0.609 for criterion 9, and the EWM weight of 0.040 obtained by the kestrel was much less than the mandarin duck’s EWM weight of 0.147, thus placing the mandarin duck in 10th place and the common kestrel in 39th place.

Finally, after optimization based on the least-squares combination, the mandarin duck ranked 10th and the kestrel ranked 19th. In general, species that are stable in their local ecosystem distribution and that can act as umbrella species should be given priority consideration as flagship species [55]. According to the statistics, of the nine flagship species other than the mandarin duck, 88.9% had more stable populations and distributions in Lishui each year, and to some extent could cover Lishui birds at different times of the year. However, only 44.4% of the nine flagship species and 6.1% of the three hundred and sixty-one birds were well known to the public. Flagship species are those that capture the imagination of the public and induce support for conservation action, and the level of public familiarity with the species greatly influences conservation efforts [56,57]. A comprehensive comparison therefore suggests that the mandarin duck is more suitable than the common kestrel as a flagship species in Lishui.

#### 4.1.2. Species Suitability Distribution Area Range Prediction

Although a variety of species distribution models have been developed to predict distributions, most species distribution models have certain requirements for data sources [58] and have major limitations. MaxEnt better bridges this gap. Studies have shown that the MaxEnt model outperforms other models in terms of prediction accuracy, especially when the amount of species distribution data is limited and only the presence point data of species can be obtained [59,60]. The MaxEnt model was thus chosen to predict the potential distribution of the species in this study [61]. The current research using the MaxEnt model mainly considers the influence of natural factors such as climate and altitude, and this method is not suitable for areas that are increasingly affected by human activities [62,63]. However, in recent years, human activities have seriously affected the distribution habitats of species. Birds, as intermediate and high-level consumers in the food chain, are highly sensitive to environmental changes. Their number and distribution characteristics play an important role in indicating the status of other members of the ecosystem [64]. Therefore, based on the background survey data of birds in Lishui, this paper used the MaxEnt model to simulate the spatial distribution patterns of birds in Lishui considering both natural and human factors. The results showed that the MaxEnt model did perform well, and the results were easy to understand. The findings indicated that the MaxEnt model could be used to provide a scientific basis for biodiversity conservation planning.

#### 4.1.3. The Shortcomings of the Method

The rating criteria and determined scores formulated in the AHP-EWM method mainly summarize the opinions of some experts and publications, however, the opinions from public was missing. Whether a flagship species can fully effectived depends largely on public support [65]. Moreover, the scoring results of this method are based on existing conditions and have a certain timeliness. Therefore, in the future, the scoring results of species may change with time, and it is necessary to regularly detect whether the selected flagship species are still applicable.

The composition and structure of biological communities are largely influenced by the environmental factors shaping the habitat [66]. The habitat and viability of species will change over time [67]. Temporal changes in suitable habitats for different species may occur at different rates [68]. Future work should regularly monitor the role of these selected flagship species in biodiversity conservation in Lishui and adjust the number and species of flagship species that are not suitable for biodiversity conservation on a regular basis.

### 4.2. Effectiveness of Flagship Species in Playing a Conservation Leadership Role

Flagship species are one of the most common marketing tools for biodiversity conservation [69]. Natural resource conservationists often use flagship species to raise conservation funds, stimulate positive conservation attitudes toward the species, and raise awareness to reduce biodiversity loss [12]. Few marketing tools have been as effective in rallying support for conservation as those based on flagship species, where each species is the focus of conservation marketing campaigns based on the characteristics it possesses that appeal to its target audience [70]. The public preference for charismatic bird and mammal species is reflected in a greater willingness to pay for their conservation [71]. Flagship species can play the expected role of target species in conservation planning for wildlife reserves in China [23].

Birds are one of the most well-known and popular groups of animals [72,73,74]. The 10 flagship species selected in this study were not only protected species but also had the characteristics of gorgeous plumage, pleasant calls, and lovely appearances [75]. These species covered land birds, swimming birds, raptors, songbirds and climbing birds. The use of multi-species strategies can usually expand the biodiversity coverage under the umbrell and flagship species conservation method, thereby better representing the habitat requirements of sympatric species [23]. From a single point of view, the suitable distribution area of any of the 10 selected flagship species could cover 30.47–78.67% of Lishui’s 361 birds, but the total suitable distribution areas of the 10 flagship species could cover all of the 361 birds surveyed in Lishui. In addition, their adaptive distribution areas covered most of the key areas of biodiversity distribution in Lishui. Therefore, habitat protection for the suitable distribution areas of these 10 flagship species is also conducive to the protection of all of the birds in the city, and protecting these species could serve as umbrella protection. In addition to representing different aspects of biodiversity, the identification of a single umbrella species is inevitably limited by factors such as randomness, demography, phenology, and sampling efforts. Thus, biodiversity hotspots may not be occupied by a single umbrella species, thus providing an additional argument for the selection of multiple umbrella flagship species [76].

Increasing the protection of flagship species can effectively promote the protection of other threatened non-flagship species and their habitats and ecosystems [77]. Some species are closely related to local cultures. The publicity and promotion of these charming flagship species are conducive to promoting scientific education on biodiversity conservation and the importance of biodiversity conservation to humans and natural ecosystems. In addition, the promotion of these flagship species could help to attract the attention of the government, social groups, and the public to the cause of biodiversity conservation, increase the influence of urban biodiversity, and stimulate public awareness and action for conservation [78,79]. Governments, businesses, and individuals should be encouraged to raise more financial support for flagship species populations, habitats, and other conservation projects [15,80], thereby increasing capital investment in urban biodiversity conservation. The promotion of flagship species is also conducive to the continuous improvement of biodiversity data management and monitoring platforms and the use of policy advantages and competitive incomes to attract more talented people with expertise to join local biodiversity conservation efforts [81]. Finally, these efforts will result in the holistic and comprehensive protection of local biodiversity [5,82] and achieve the construction of efficient protection management systems.

### 4.3. Challenges and Recommendations for Local Governments in Using Flagship Species

The challenges faced by local governments in flagship species protection are multifaceted, including the requirements for funds, professional knowledge, and skills, as well as interference caused by human activities [83,84]. Local governments need to strengthen the promotion and publicity of flagship species and tap into their cultural and ornamental values. The advertising effects of flagship species is continuously strengthened through commercial or public interest promotion [85]. Flagship species can also provide product endorsements for local food, cultural and creative events and enterprises, and for commercial companies [86], enriching product content and enhancing brand images, or even through the registration of the flagship species image as a trademark [87]. Biodiversity products such as art, literature, and music could also be developed around flagship species to raise awareness and expand the city’s influence from a cultural perspective. Local governments also can develop ecotourism led by flagship species. Flagship species publicity and promotion are conducive to improving public awareness of flagship species and encouraging people to participate in relevant ecotourism, thereby improving local finance and local incomes.

## 5. Conclusions

Based on the survey results of bird diversity in Lishui, this study used a combination of AHP, EWM, AHP-EWM, and MaxEnt to select out 10 bird species that can both serve as flagship species and umbrella species. It is hoped that the flagship species of Lishui will play a leading role in the protection of Lishui’s biodiversity and provide a reference for the selecting of flagship species at the municipal local level.

## Figures and Tables

**Figure 1 animals-13-01825-f001:**
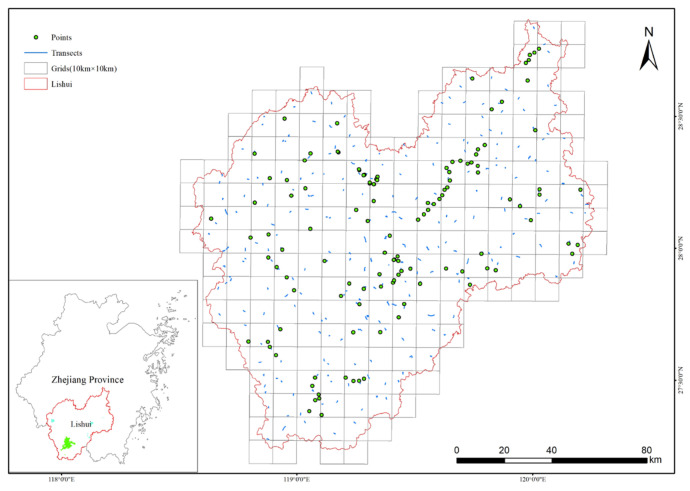
The geographical location of Lishui, Zhejiang province, China, and the distribution of bird survey transects and points in Lishui.

**Figure 2 animals-13-01825-f002:**
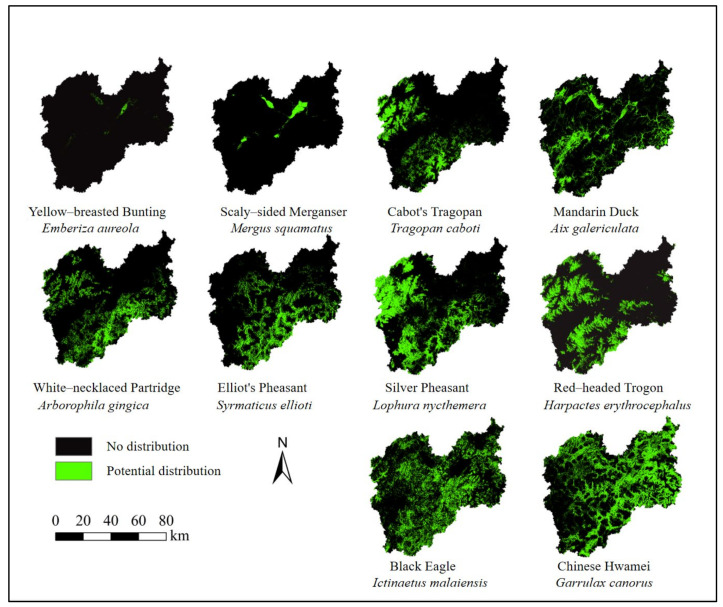
The potential distribution of the top 10 species.

**Figure 3 animals-13-01825-f003:**
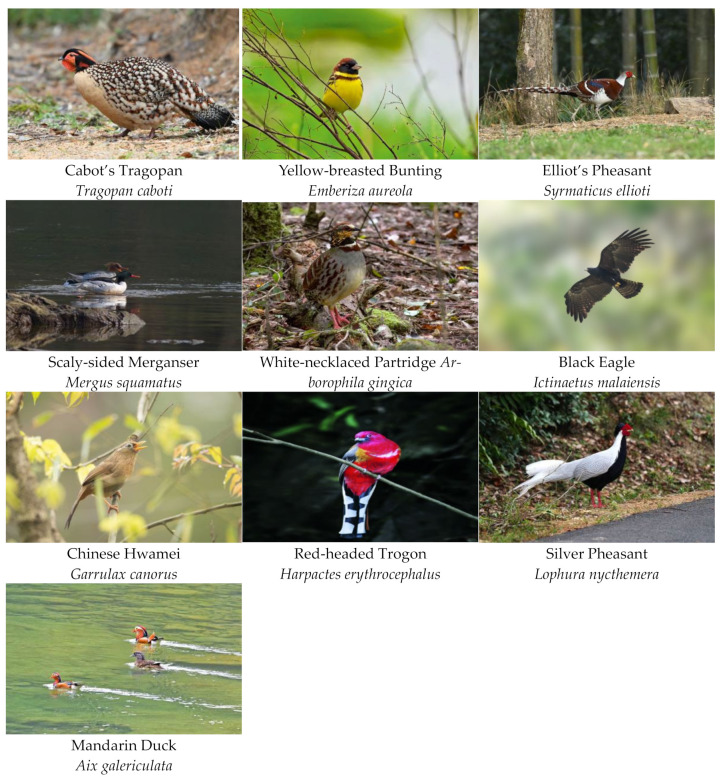
The 10 flagship species selected in Lishui, Zhejiang Province, China were not only rare protected species but also had the characteristics of gorgeous plumage, pleasant calls, and lovely appearances.

**Figure 4 animals-13-01825-f004:**
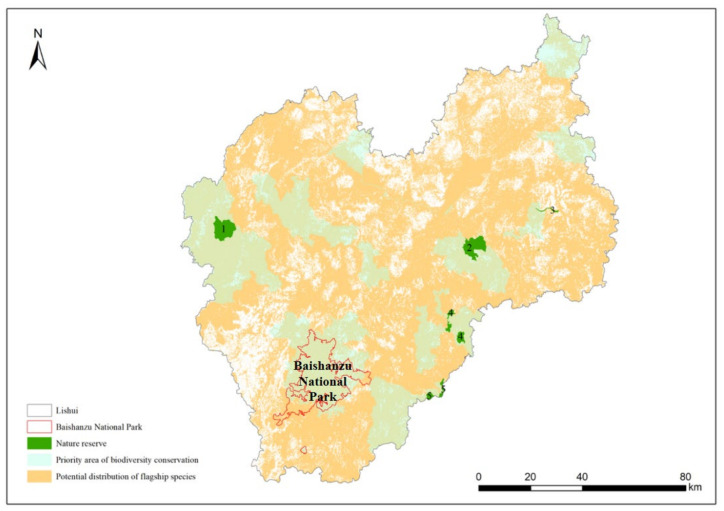
Results of overlaying flagship species’ suitable distribution areas with protected areas in Lishui. 1: Zhejiang Jiulongshan National Nature Reserve; 2: Liandu Fengyuan Provincial Nature Reserve; 3: Qingtian *Pelochelys cantorii* Provincial Nature Reserve; 4: Jingning Dayang lake wetland group Provincial Nature Reserve; 5: Jingning Wangdongyang alpine wetland Provincial Nature Reserve.

**Table 1 animals-13-01825-t001:** Environmental data (climate, land use/cover, topography and elevation, and man-made interference) source table.

Environment Variable	Data Sources
Climate	Climate data were downloaded from the Global Climate Data Center (http://www.wordclim.org/, (accessed on 21 January 2022)).
Land use/cover	Land use/cover data were obtained from GlobeLand30: Global Geographic Information Public Product (http://www.globallandcover.com/, (accessed on 21 January 2022)).
Topography and elevation	Topographic and elevation data were obtained from digital elevation model (DEM) data with a resolution of 30 m and downloaded from the Geospatial Data Cloud, and slope and aspect data were extracted from DEM data using ArcGIS 10.7 software. Normalized difference vegetation index (NDVI) data were downloaded from the National Ecological Science Data Center’s 30 m annual maximum NDVI dataset for China from 2000 to 2020.
Man-made interference	Population density data were downloaded from China’s 1 km population density data for 2020 (http://www.worldpop.org/, (accessed on 21 January 2022)). The road data were obtained from the (http://openstreetmap.org/, (accessed on 21 January 2022)), and the distances from roads and settlements were obtained by calculating the Euclidean distances for roads as well as settlements using ArcGIS 10.7 software.

**Table 2 animals-13-01825-t002:** Evaluation basis and reference for flagship species selection factors.

No.	Element Name	Criteria
1	Conservation grading	Whether the target species was listed as a Class I or II state key protected wild animal. They were classified into three levels. Class I state key protected animal: 1. Class II state key protected animal: 0.6. Non-protected animal: 0.1.
2	Endangered status	According to the evaluation criteria of the International Union for Conservation of Nature, endangered species were divided into seven levels. Extinct and extinct in the wild were removed from the list without matching objects. They were classified into seven levels. Critically endangered: 1. Endangered: 0.8. Vulnerable: 0.6. Near threatened: 0.2. Least concern: 0.1.
3	Stability of population size and distribution	Whether the target species had a stable distribution in Lishui every year. They were classified into three levels. The population size and distribution were relatively stable and common: 1. The population size and distribution were relatively stable and rare: 0.6. The population size and distribution were unstable and rare: 0.
4	Research of population size and distribution	Whether the distribution areas and size of bird populations in Lishui were already well known. They were classified into three levels. Fully understood: 1. Not well understood: 0.6. Not understood: 0.1.
5	Socio-economic values	Whether the target species had the main value or potential value to promote bird watching and other ecotourism. They were classified into three levels. High: 1. Middle: 0.6. Low: 0.1.
6	Cultural values	The relationship between the target species and local culture, including folklore, art, food, or handicrafts. They were classified into three levels. High: 1. Middle: 0.6. Low: 0.1.
7	Endemic species of China	Whether the target species was endemic to China. They were classified into two levels. Yes: 1. No: 0.6.
8	Unique in Lishui, Zhejiang	In Zhejiang Province, whether the target species were only distributed or mainly distributed in Lishui. They were classified into two levels. Yes: 1. No: 0.6.
9	Recognition	The degree of recognition of target species by different social groups. They were classified into three levels. Known to the public: 1. Known to the biodiversity managers of a government agency: 0.6. Only known to professional bird researchers: 0.1.

**Table 3 animals-13-01825-t003:** Top 50 species ranked by the analytic hierarchy process (AHP), the entropy weight method (EWM) and the analytic hierarchy process-entropy weight method (AHP-EWM).

English Name	Scientific Name	AHP	EWM	AHP-EWM
Cabot’s Tragopan	*Tragopan caboti*	0.962	0.969	0.958
Yellow-breasted Bunting	*Emberiza aureola*	0.958	0.710	0.866
Elliot’s Pheasant	*Syrmaticus ellioti*	0.905	0.815	0.860
Scaly-sided Merganser	*Mergus squamatus*	0.874	0.795	0.843
White-necklaced Partridge	*Arborophila gingica*	0.784	0.759	0.766
Black Eagle	*Ictinaetus malaiensis*	0.724	0.671	0.707
Chinese Hwamei	*Garrulax canorus*	0.699	0.672	0.664
Red-headed Trogon	*Harpactes erythrocephalus*	0.657	0.675	0.655
Silver Pheasant	*Lophura nycthemera*	0.679	0.657	0.643
Mandarin Duck	*Aix galericulata*	0.636	0.656	0.625
Crested Serpent Eagle	*Spilornis cheela*	0.647	0.610	0.624
Blue-throated Bee-eater	*Merops viridis*	0.616	0.641	0.604
Fujian Niltava	*Niltava davidi*	0.628	0.586	0.603
Koklass Pheasant	*Pucrasia macrolopha*	0.628	0.586	0.603
Buffy Laughingthrush	*Pterorhinus berthemyi*	0.656	0.585	0.598
Mountain Hawk-Eagle	*Nisaetus nipalensis*	0.637	0.537	0.592
Amur Falcon	*Falco amurensis*	0.637	0.537	0.592
Crested Goshawk	*Accipiter trivirgatus*	0.637	0.537	0.592
Common Kestrel	*Falco tinnunculus*	0.662	0.525	0.585
Eurasian Sparrowhawk	*Accipiter nisus*	0.618	0.522	0.571
Black Kite	*Milvus migrans*	0.618	0.522	0.571
Black Baza	*Aviceda leuphotes*	0.618	0.522	0.571
Chinese Sparrowhawk	*Accipiter soloensis*	0.618	0.522	0.571
White-throated Kingfisher	*Halcyon smyrnensis*	0.618	0.522	0.571
Red-billed Leiothrix	*Leiothrix lutea*	0.588	0.572	0.566
Asian Barred Owlet	*Glaucidium cuculoides*	0.587	0.581	0.557
Peregrine Falcon	*Falco peregrinus*	0.575	0.521	0.553
Grey-faced Buzzard	*Butastur indicus*	0.575	0.521	0.553
Merlin	*Falco columbarius*	0.575	0.521	0.553
Black-winged Kite	*Elanus caeruleus*	0.575	0.521	0.553
Crested Honey Buzzard	*Pernis ptilorhynchus*	0.575	0.521	0.553
Western Osprey	*Pandion haliaetus*	0.575	0.521	0.553
Northern Goshawk	*Accipiter gentilis*	0.575	0.521	0.553
Eastern marsh Harrier	*Circus spilonotus*	0.575	0.521	0.553
Eastern Grass Owl	*Tyto longimembris*	0.599	0.538	0.552
Brown Wood Owl	*Strix leptogrammica*	0.600	0.529	0.552
Eurasian Hobby	*Falco subbuteo*	0.555	0.505	0.532
Besra	*Accipiter virgatus*	0.555	0.505	0.532
Japanese Sparrowhawk	*Accipiter gularis*	0.555	0.505	0.532
Eastern Buzzard	*Buteo japonicus*	0.555	0.505	0.532
Siberian Rubythroat	*Calliope calliope*	0.555	0.505	0.532
Common Kingfisher	*Alcedo atthis*	0.526	0.597	0.526
Short-tailed Parrotbill	*Neosuthora davidiana*	0.589	0.464	0.520
Eurasian Eagle-Owl	*Bubo bubo*	0.589	0.464	0.520
Barred Cuckoo-Dove	*Macropygia unchall*	0.537	0.513	0.513
Eurasian Hoopoe	*Upupa epops*	0.482	0.594	0.512
Sultan Tit	*Melanochlora sultanea*	0.495	0.542	0.507
Collared Owlet	*Taenioptynx brodiei*	0.570	0.449	0.499
Collared Scops Owl	*Otus lettia*	0.570	0.449	0.499
Oriental Scops Owl	*Otus sunia*	0.570	0.449	0.499

**Table 4 animals-13-01825-t004:** The receiver operating characteristics (ROC) analysis was used to test the area under the curve (AUC) values obtained from the predicted results of MaxEnt in the top 10 birds.

English Name	Scientific Name	Number of Species Covered	Percentage (%)	AUC
Yellow-breasted Bunting	*Emberiza aureola*	110	30.47	0.996
Scaly-sided Merganser	*Mergus squamatus*	274	75.90	0.989
Cabot’s Tragopan	*Tragopan caboti*	147	40.72	0.940
Mandarin Duck	*Aix galericulata*	284	78.67	0.916
White-necklaced Partridge	*Arborophila gingica*	186	51.52	0.915
Elliot’s Pheasant	*Syrmaticus ellioti*	189	52.35	0.904
Silver Pheasant	*Lophura nycthemera*	208	57.62	0.896
Red-headed Trogon	*Harpactes erythrocephalus*	177	49.03	0.893
Black Eagle	*Ictinaetus malaiensis*	203	56.23	0.821
Chinese Hwamei	*Garrulax canorus*	253	70.08	0.794
Total		361	100.00	

**Table 5 animals-13-01825-t005:** National protection level, IUCN endangerment categories, endemic to China status, duration of presence in the area, and ecological niche of 10 flagship species.

English Name	Scientific Name	National Protection Level	IUCN Endangerment Categories	Endemic to China	Duration of Presence in the Area	Ecological Type
Cabot’s Tragopan	*Tragopan caboti*	I	EN	√	residents	land bird
Yellow-breasted Bunting	*Emberiza aureola*	I	CR		passage migrants	songbird
Elliot’s Pheasant	*Syrmaticus ellioti*	I	VU	√	residents	land bird
Scaly-sided Merganser	*Mergus squamatus*	I	EN		winter birds	swimming birds
White-necklaced Partridge	*Arborophila gingica*	II	VU	√	residents	land bird
Black Eagle	*Ictinaetus malaiensis*	II	VU		residents	raptor
Chinese Hwamei	*Garrulax canorus*	II	NT		residents	songbird
Red-headed Trogon	*Harpactes erythrocephalus*	II	NT		residents	climbing birds
Silver Pheasant	*Lophura nycthemera*	II	LC		residents	land bird
Mandarin Duck	*Aix galericulata*	II	NT		winter birds	swimming birds

## Data Availability

The data are already available with the required references provided in the paper.

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
