# Peer review of "Selection of Flagship Species and Their Use as Umbrellas in Bird Conservation: A Case Study in Lishui, Zhejiang Province, China"

_animals, 2023, doi:10.3390/ani13111825_

Round 1

Reviewer 1 Report

Please see the attached MS Word file.

Author Response

Response to Reviewer 1 Comments

Dear Editors and Reviewers:

Thank you for your letter and for the reviewer's comments concerning our manuscript entitled "Selection of flagship species and their use as umbrellas in bird conservation: a case study in Lishui, Zhejiang Province, China" (ID: 2394122). Those comments are all valuable and very helpful for revising and improving our paper, as well as the important guiding significance to our researches. We have studied comments carefully and have made correction which we hope meet with approval. Revised portion are marked in red in the paper. The main corrections in the paper and the responds to the reviewer's comments are as flowing:

Responses to the comments of Reviewer #1

Point 1: Line 102-103: Abies beshanzuensis in italics

Response 1: We have changed "Abies beshanzuensis" to italics "Abies beshanzuensis".

Point 2: Line 114: Flora, lower case f

Response 2: "Flora" has been changed to lower case f "flora".

Point 3: Line 98 and Lines 133-143: The legal status of the study area is not clearly described. Line 98 introduces the name Lishui without saying what it is. Line 134 says that Lishui is a city, but the rest of the description makes it clear that much surrounding natural habitat is included. So the name Lishui seems to refer to both the city and the administrative district (county? prefecture?). Are there separate protected areas (national or provincial) within this? There could be a simple cross-reference to section 3.3 in the Results. This has some implications for other parts of the text. For example, line 421-422 says that the results are "conducive to protection of all birds in the city", but readers will not think of a nature reserve as being a "city".

Response 3: As suggested by the reviewer, we have a clearer description of Lishui City in ' Line 98, Line 133-143 and Section 3.3 ', and added ' Zhejiang Province ' to the title, hoping to help readers understand that Lishui is a prefecture-level city in Zhejiang Province.

Point 4: Table 1: I had some questions regarding the choice and significance of the criteria. Criterion 2 (level of endangerment) scores rare species high, but criterion 2 (population size) scores common species high. I could not find a statement on how population size had been assessed (perhaps I overlooked it), leaving me in doubt whether the meaning is global, national or local population size. I assume some criteria (socio-economic value, cultural value, ease of recognition) were based on pooled expert opinion of the authors rather than social surveys-a valid method, but it would be nice to state this. As socio-economic value refers to tourism potential, it ignores other such values (ecological roles, pollination, seed dispersal, pest control).

Response 4: Considering the Reviewer's suggestion, I think reviewer mean that we have a higher score for rare species in standard 2 (endangered), and a higher score for common species in standard 4 (population size). Here may be some misunderstandings caused by our lack of English expression ability. Standard 4 we want to express is the evaluation of population size knowability, we already know that the population size of the species score is high, otherwise the score is low. We have changed the description of the standard 4 in Table 1 into "Whether the distribution areas and size of bird populations in Lishui were already well known. They were classified into 3 levels. Fully understand: 1. Not well understood: 0.6. Don't understand: 0.1." Hoping to express our original intention.

As Reviewer suggested that we have added "These criteria and the determination of scores are based on the combined opinions of experts and authors in the relevant fields, rather than social surveys." in the article.

We consider that criteria 5 and 6, in addition to others, can demonstrate the ecological value of flagship species.

Point 5: Line 232: A better term for "traveller birds" may be passage migrants (i.e., those travelling through the area without residing there for a sustained period).

Response 5: As Reviewer suggested that the statements of "traveller birds" were corrected as "passage migrants".

Point 6: Lines 255-260: Inconsistency in use of upper and lower case. Within a few lines text we have mandarin duck, Mandarin Duck, white-necklaced partridge, White-necklaced Partridge, etc.

Response 6: We are very sorry for our negligence of inconsistency in use of upper and lower case. We have standardized the case of the English names of birds to upper case.

Point 7: Lines 499-508: I have a different opinion on whether laws and regulations are suitable for enforcing management of flagship species, but perhaps this differs from country to country. If the criteria used to identify flagship species include features such as population size, flagship species themselves might change. However, this is not a criticism of the paper, just an observation.

Response 7: Thanks to the Reviewer's advice, but after discussion we believe that flagship species may be poached by humans due to their outstanding economic, cultural and ecological values, so we hope that the state can formulate relevant laws to manage the protection of flagship species to prevent their abnormal decline. Regarding the use of flagship species population sizes as identification, we similarly suggest that flagship species themselves may be subject to change. But the selection of flagship species should be time-limited, not permanent. Based on changes in the characteristics of the flagship species itself, we should judge in time whether it still qualifies as a flagship species.

We tried our best to improve the manuscript and made some changes in the manuscript. These changes will not influence the content and framework of the paper. And here we did not list the changes but marked in red in revised paper.

We appreciate for editor's and reviewer's warm work earnestly, and hope that the correction will meet with approval.

Once again, thank you very much for your comments and suggestions.

Best regards!

Yours sincerely,

Yifei Wang

Reviewer 2 Report

Comments to Authors

Title: Selection of Flagship Species and Their Use As Umbrellas in 2 Regional Bird Conservation: A Case Study in Lishui, China

The authors using the technological tools with data on bird species distribution come out with constructive conservation planning for biodiversity conservation following the flagship or umbrella species concept. Data from such study highlights the importance of preserving the existing or enlarging the Protected Areas and upgrading the Nature Reserves into Protected Areas. Thus, the paper is deserving for publication. However, the manuscript requires minor editing and changes as listed below including language editing and if these are incorporated the manuscript will improve in clarity. 

Line No. 30-31: The distribution area of these 10 species covered the nature reserves and the priority area of biodiversity of Lishui, in that these 10 species can also serve as umbrella species to protect local biodiversity.

In addition to the above, mention what are the niches [nectivore, frugivore, insectivore, etc.] these species use? Because if 10 species represent diverse niche, it is understood efforts taken for successful conservation of these 10 species would ensure conservation of the entire biodiversity. 

Line No. 39-40: The Earth is currently experiencing its sixth mass extinction [1,2] due to climate 39 change, overexploitation, environmental change and habitat destruction [3].

In the above, I suggest the authors to list the causes in an orderly maner that took place like habitat destruction, overexploitation, environmental change and climate change.

Line No. 50-51: It is essential to choose the most effective…..

In the above please add at the beginning of the statment, ‘Given the scenario’,  and replace the word ‘most’ with ‘cost’ 

Line No. 54-55: Among these approaches, the use of flagship species and umbrella species has become an extremely valuable strategy for restoration and biodiversity conservation.

Please delete the word species that is after the falgship and replace the words ‘has become an extremely valuable strategy for restoration and biodiversity conservation’ with ‘has been an effective and widely used concept for restoration and conservation of biodiversity across the globe’

Line No. 91-92: A survey revealed the presence of 42 species of rare and endangered plants among …

Modify the above as: A recent botanical survey revealed the presence of 42 species of rare and endangered plants among.

Line No. 96-98: Although some wildlife and ecosystem conservation initiatives have been undertaken in Lishui, the current state of survival of many wildlife populations and the current state of ecosystem fragmentation remain unclear.

In the above statement saying ‘the current state of survival of many wildlife populations and the current state of ecosystem fragmentation remain unclear’ is either weekening your recommendations or showing that you are suggesting the new concept, without understanding what the current conservation issues are?, which is meaningless because recommendations should be based on what the issues are.

To overcome the lacuna, authors are suggested to highlight what are the conservation issues that threaten the biodiversity conservation in the area concenrned by reviewing some of the existing literatures.

Line No. 108-109: This study selected birds as the target population. Based on results of a background survey of bird diversity in the city of Lishui …..

Modify above as This study selected birds as the target species based on results of a background survey on bird diversity in the city of Lishui (mention the reference here). And then break the rest of the statement into a new sentnce.

Line No. 112-113: The flagship species 112 concept could play a leading role in conservation and provide guidance for biodiversity 113 conservation in the city.

Delete the above sentence, as importance of the concept is already discussed.   

Line 118: Lishui is the prefecture-level city with…

In the above change perfecture-level into perfect city

Liine No. 122: southwest to the northeast. Please give a reference here.

Figure 1. The location of transects in Lishui.

Modify the above legend as ‘Map showing the study area Lishui with layout of transect locations’

Also make the line transects as straight lines. The line transect maked in the map are looks like a mix of lines and dots.

Is the map showing the locations of 112 point counts and if not authors should also show the locations with solid dots.

The map should also show Baishanzu National Park, five nature reserves at or above the provincial level

Line No. 131: The bird distribution data were derived from the background survey of bird diversity in Lishui. Please mention the reference the background survey here.

Line No. 137: In this study, 27 environmental variables affecting species distribution were selected

In the above using word affecting indcate negative term. Is it so that only enviromental factors that affect the birds species negatively alone selected and if so why? And if not change the word ‘affecting the species’ with ‘infleuncing the bird species’

The various data sources listed in under ‘2.3. Environmental data’ paragraph could be connverted into a table and placed in supplementary materials. So that the entire pargraph could be converted into a single satement.

Line No. 198-199: The results of MaxEnt were then superimposed on the distribution patterns of 'all 198 birds' (the 361 species modelled) to determine whether the suitable range for the species 199 included other bird sites and to select flagship species.

The above sentence is not clear. Please check the following statement is the one what the authrors try to convey here

‘The results of MaxEnt distribution model developed for falgship species were superimposed on the distribution patterns of 'all birds' (the 361 species modelled) to determine whether the suitable range of falgship species included other bird sites.’

If so, you may replace your sentece with the above sentece. Or else rewrite it clearly.

Table 4. The information of 10 flagship species.

Table title should be stand alone and thus suggest authors to elaborate all table and figure title adequately.

Further the terms used for column labeling and inside the table are not the standard one. Suggest the authros to use statndard terms for all these. Residency Type (Duration of presence in the area), Ecological Type (Eological Niche Used).

Figure 4. Results of overlaying flagship species suitable distribution areas with protected areas in 276 Lishui.

Please lable the Baishanzu National Park and the five nature reserves. 

In the above map, the priority area for Biodiversity coservation is not clearly visible in the map. Authros must give a colour that show them prominantly.

In the Discussion after highlighting the what is the uniquness of the study findings in improving biodiversity conservation, what are the lacunae need to be discussed at the end. Therefore the authors should take first two sub-heading to later stage of the discussion.  

Discussion about Public education & Wildlife Tourism too much and this study has not evaluated about these and thus the discussion on these aspects need to be cutshort.

Minor editing of English language

Author Response

Response to Reviewer 2 Comments

Dear Editors and Reviewers:

Thank you for your letter and for the reviewer's comments concerning our manuscript entitled "Selection of flagship species and their use as umbrellas in bird conservation: a case study in Lishui, Zhejiang Province, China" (ID: 2394122). Those comments are all valuable and very helpful for revising and improving our paper, as well as the important guiding significance to our researches. We have studied comments carefully and have made correction which we hope meet with approval. Revised portion are marked in red in the paper. The main corrections in the paper and the responds to the reviewer's comments are as flowing:

Responses to the comments of Reviewer #2

Point 1: Line No. 30-31: The distribution area of these 10 species covered the nature reserves and the priority area of biodiversity of Lishui, in that these 10 species can also serve as umbrella species to protect local biodiversity. In addition to the above, mention what are the niches [nectivore, frugivore, insectivore, etc.] these species use? Because if 10 species represent diverse niche, it is understood efforts taken for successful conservation of these 10 species would ensure conservation of the entire biodiversity.

Response 1: As Reviewer suggested, we decided to increase the description of ecological types after discussion. Because in the following we also have a summary of the ecological types. We believe that ecotypes can also play an ecological role in niche proximity. "These 10 species contain five ecological niches (land birds, songbirds, swimming birds, raptors and climbing birds), and their distribution areas cover the nature reserves and the priority area of biodiversity of Lishui, in that these 10 species can also serve as umbrella species to protect local biodiversity."

Point 2: Line No. 39-40: The Earth is currently experiencing its sixth mass extinction [1,2] due to climate 39 change, over-exploitation, environmental change and habitat destruction [3].In the above, I suggest the authors to list the causes in an orderly manner that took place like habitat destruction, over-exploitation, environmental change and climate change.

Response 2: We revised it according to the opinions of reviewers and reordered it as "habitat destruction, over-exploitation, environmental change, climate change and other reasons".

Point 3: Line No. 50-51: It is essential to choose the most effective….. In the above please add at the beginning of the statement, 'Given the scenario', and replace the word 'most' with 'cost'

Response 3: As Reviewer suggested that we have added "In the case of limited capital supply and protection" at the beginning and have replaced the word 'most' with 'cost'.

Point 4: Line No. 54-55: Among these approaches, the use of flagship species and umbrella species has become an extremely valuable strategy for restoration and biodiversity conservation. Please delete the word species that is after the flagship and replace the words 'has become an extremely valuable strategy for restoration and biodiversity conservation' with 'has been an effective and widely used concept for restoration and conservation of biodiversity across the globe'.

Response 4: We have made correction according to the Reviewer's comments.We have deleted the word 'species' after 'flagship' and replaced the words 'has become an extremely valuable strategy for restoration and biodiversity conservation' with 'has been an effective and widely used concept for restoration and conservation of biodiversity across the globe'.

Point 5: Line No. 91-92: A survey revealed the presence of 42 species of rare and endangered plants among …Modify the above as: A recent botanical survey revealed the presence of 42 species of rare and endangered plants among.

Response 5: As Reviewer suggested that we have modified the original sentence to "A recent botanical survey revealed the presence of 42 species of rare and endangered plants among."

Point 6: Line No. 96-98: Although some wildlife and ecosystem conservation initiatives have been undertaken in Lishui, the current state of survival of many wildlife populations and the current state of ecosystem fragmentation remain unclear. In the above statement saying 'the current state of survival of many wildlife populations and the current state of ecosystem fragmentation remain unclear' is either weakening your recommendations or showing that you are suggesting the new concept, without understanding what the current conservation issues are? which is meaningless because recommendations should be based on what the issues are. To overcome the lacuna, authors are suggested to highlight what are the conservation issues that threaten the biodiversity conservation in the area concerned by reviewing some of the existing literatures.

Response 6: As Reviewer suggested that We have deleted this sentence and simplified it.

Point 7: Line No. 108-109: This study selected birds as the target population. Based on results of a background survey of bird diversity in the city of Lishui …..Modify above as This study selected birds as the target species based on results of a background survey on bird diversity in the city of Lishui (mention the reference here). And then break the rest of the statement into a new sentence.

Response 7: We have made correction according to the Reviewer's comments. We have broken it down into two sentences:"This study selected birds as the target species based on results of a background survey on bird diversity in the city of Lishui." and "In this study, we used the analytic hierarchy process (AHP)……"

Point 8: Line No. 112-113: The flagship species concept could play a leading role in conservation and provide guidance for biodiversity conservation in the city. Delete the above sentence, as importance of the concept is already discussed.

Response 8: It is really true as Reviewer suggested that we have deleted "The flagship species concept could play a leading role in conservation and provide guidance for biodiversity conservation in the city."

Point 9: Line 118: Lishui is the prefecture-level city with…In the above change perfecture-level into perfect city

Response 9: I am sorry that this part was not clear in the original manuscript. I should have explained that we want to convey that "Lishui" is a "prefecture-level city" (China 's classification of cities) and not to describe its beauty or not.

Point 10: Line No. 122: southwest to the northeast. Please give a reference here.

Response 10: As Reviewer suggested that we have added the reference "Yao, J.J. Studies on the urban plant biodiversity in Lishui city Zhejiang province. Huazhong Agricultural University 2009." here.

Point 11: Figure 1.: The location of transects in Lishui. Modify the above legend as 'Map showing the study area Lishui with layout of transect locations' Also make the line transects as straight lines. The line transect maked in the map are looks like a mix of lines and dots. Is the map showing the locations of 112 point counts and if not authors should also show the locations with solid dots. The map should also show Baishanzu National Park, five nature reserves at or above the provincial level.

Response 11: We have made correction according to the Reviewer's comments. We have changed the graphics according to the Reviewer's requirements. Due to the large area of Lishui, the sample line is relatively short on the figure, but it is indeed a line.

Point 12: Line No. 131: The bird distribution data were derived from the background survey of bird diversity in Lishui. Please mention the reference the background survey here.

Response 12: Thanks for the Reviewer's valuable advice, but the distribution data of Lishui birds in this article are all collected by our team. We use raw data for analysis without citing literature.

Point 13: Line No. 137: In this study, 27 environmental variables affecting species distribution were selected In the above using word affecting indicate negative term. Is it so that only environmental factors that affect the birds species negatively alone selected and if so why? And if not change the word 'affecting the species' with 'influencing the bird species' The various data sources listed in under ' 2.3. Environmental data' paragraph could be converted into a table and placed in supplementary materials. So that the entire paragraph could be converted into a single statement.

Response 13: We have made correction according to the Reviewer's comments. We have changed 'affecting the species' with 'influencing the bird species' and converted the various data sources in 2.3 into tabular form.

Point 14: Line No. 198-199: The results of MaxEnt were then superimposed on the distribution patterns of 'all 198 birds' (the 361 species modelled) to determine whether the suitable range for the species 199 included other bird sites and to select flagship species. The above sentence is not clear. Please check the following statement is the one what the authors try to convey here 'The results of MaxEnt distribution model developed for flagship species were superimposed on the distribution patterns of 'all birds' (the 361 species modelled) to determine whether the suitable range of flagship species included other bird sites.' If so, you may replace your sentence with the above sentence. Or else rewrite it clearly.

Response 14: Thanks for your comments. We have changed the original sentence to:'The results of MaxEnt distribution model developed for flagship species were superimposed on the distribution patterns of 'all birds' (the 361 species modelled) to determine whether the suitable range of flagship species included other bird sites.'

Point 15: Table 4. The information of 10 flagship species. Table title should be stand alone and thus suggest authors to elaborate all table and figure title adequately. Further the terms used for column labeling and inside the table are not the standard one. Suggest the authors to use standard terms for all these. Residency Type (Duration of presence in the area), Ecological Type (Ecological Niche Used).

Response 15: We have made correction according to the Reviewer's comments. We have changed the title to "National protection level, IUCN endangerment categories, Endemic to China, Duration of presence in the area and Ecological Niche of 10 flagship species." We have changed the "Residency Type" to "Duration of presence in the area". But after discussion and thinking, we believe that the "Ecological Type" may be more appropriate.

Point 16: Figure 4. Results of overlaying flagship species suitable distribution areas with protected areas in 276 Lishui. Please liable the Baishanzu National Park and the five nature reserves. In the above map, the priority area for Biodiversity conservation is not clearly visible in the map. Authors must give a color that show them prominently.

Response 16: As Reviewer suggested that we have adjusted the colors of Baishanzu National Park and five nature reserves to make them more visible and clearly visible.

Point 17: In the Discussion after highlighting the what is the uniqueness of the study findings in improving biodiversity conservation, what are the lacunae need to be discussed at the end. Therefore, the authors should take first two sub-heading to later stage of the discussion.

Response 17: Thank you very much for the Reviewer's suggestion. We have rearrange the discussion according to the reviewer's comments.

Point 18: Discussion about Public education & Wildlife Tourism too much and this study has not evaluated about these and thus the discussion on these aspects need to be cut short.

Response 18: It is really true as Reviewer suggested that we have reduced the two sections of public education and wildlife tourism in the discussion.

We tried our best to improve the manuscript and made some changes in the manuscript. These changes will not influence the content and framework of the paper. And here we did not list the changes but marked in red in revised paper.

We appreciate for editor's and reviewer's warm work earnestly, and hope that the correction will meet with approval.

Once again, thank you very much for your comments and suggestions.

Best regards!

Yours sincerely,

Yifei Wang
